# The Supraclavicular Artery Island Flap for Pharynx Reconstruction

**DOI:** 10.3390/jcm11113126

**Published:** 2022-05-31

**Authors:** Eirini Nikolaidou, Glykeria Pantazi, Apostolos Sovatzidis, Stella Vakouli, Chrysoula Vardaxi, Iraklis Evangelopoulos, Spyridon Gougousis

**Affiliations:** 1Department of Plastic, Reconstructive and Hand Surgery & Burns ICU, General Hospital of Thessaloniki “G. Papanikolaou”, 57010 Thessaloniki, Greece; micropan@otenet.gr; 2Department of Surgery, General Hospital of Giannitsa, 58100 Pella, Greece; t.sovatzidis@gmail.com; 3Department of Otorhinolaryngology-Head and Neck Surgery, General Hospital of Thessaloniki “G. Papanikolaou”, 57010 Thessaloniki, Greece; stellavak@gmail.com (S.V.); vardachr@gmail.com (C.V.); 4School of Medicine, National and Kapodistrian University of Athens, 11528 Athens, Greece; iraklisevangel@gmail.com; 5E.N.T. Department General Hospital of Thessaloniki “G. Papanikolaou”, Aristotle University of Thessaloniki, 54124 Thessaloniki, Greece; spyridong@auth.gr

**Keywords:** supraclavicular artery island flap, pharynx reconstruction, review, laryngectomy

## Abstract

The supraclavicular artery island flap (SCAIF) is a reliable, easy-to-harvest and versatile fasciocutaneous flap that can be used for pharynx reconstruction. Instead of free flaps, it requires no microsurgical technique, reduced operating time and postoperative care, making it an ideal option, especially during the COVID-19 pandemic. The primary aim of our study was to present two cases of a total laryngectomy and reconstruction with the SCAIF during the pandemic. The secondary aim was to review the literature concerning surgical techniques, complications and contradictions of the SCAIF for pharynx reconstruction. A literature search was performed using the PubMed, ScienceDirect, Wiley Online Library, Google Scholar, Scopus and Cochrane Library databases, using MeSH terms: larynx AND reconstruction AND flap. Ten full-text articles comprising 92 patients with 93 supraclavicular flaps were included. The patch graft, pharyngeal interposition graft, tubularization or “U”-shaped SCAIF were the main surgical techniques. Pharyngocutaneous fistula was the most frequent postoperative complication, especially in patients with previous radiotherapy, but just 19% of patients required secondary intervention. The lack of donor-site morbidity, low flap loss rates and stenosis rates favored this reconstructive option. This review underlined that the SCAIF has comparable results with other reconstructive options, consolidating this flap in the workhorse of pharynx reconstruction.

## 1. Introduction

Pharynx reconstruction after oncologic surgery is a constant challenge. In a single operation, surgeons must ensure the integrity of the gastrointestinal tract with an acceptable aesthetic result. Το meet these two conditions, reconstruction with flaps is a necessity.

Several flaps have been described for pharynx reconstruction. The pectoralis major, latissimus dorsi and trapezius flaps are the most used regional myocutaneous flaps. Their adjacent location to the defect, in combination with the minimal operative time for experienced surgeons, makes them a preferable choice. The development of microsurgery has expanded the available reconstructive options from regional to microvascular free tissue transfer. The free radial forearm flap and the free anterolateral thigh flap have become the standard of care. Their versatility and the reliability they provide, due to their well-vascularized tissue, ensure complete coverage of the defect.

Since the outbreak of the COVID-19 pandemic, usual surgical practices have changed. The high demand for medical and paramedical staff to meet the needs of COVID-19 patients has led to significant cuts in surgical specialties. Surgeons have had to deal with reduced operative time, surgical and anesthesiological personnel and unavailability of intensive care units. Based on these needs, in March 2020, the British Association of Head and Neck Oncologists published guidelines for head and neck oncology and reconstruction [1]. According to their guidelines, surgeons should consider functional reconstruction with reduced surgical time, e.g., use of local or pedicled flaps rather than free flaps.

The supraclavicular artery island flap (SCAIF) is a thin, axial fasciocutaneous flap, based on the supraclavicular artery. It was first described by Lamberty in 1979 [2]. Initial high rates of distal flap necrosis limited its use. Twenty years later, Pallua’s anatomical studies led to a modified version of the flap [3].

The SCAIF is based on the supraclavicular artery, which originates from the transverse cervical artery. After branching 3–5 cm off the thyrocervical trunk, it pierces the deep fascia of the deltoid muscle. After a distance of 2–4.5 cm through the deep fascia, it continues its course superficially across the acromioclavicular joint [4]. The angiosome extends to the ventral surface of the deltoid muscle, up to 10 cm in width and 22 cm in length [3]. The pedicle length of the flap ranges from 1 to 7 cm. According to CT angiography studies, the mean vessel diameter is 1.5 mm [5,6]. The flap is raised distal to proximal, as described by Pallua and Demir [7]. The donor site is closed primarily for a flap width up to 8 cm.

These anatomical characteristics make the SCAIF a versatile and reliable flap for reconstruction of the neck, pharyngeal wall, tracheal, stoma defects, mandible, intraoral defects, skull base, and parotid and facial defects. The pliability of the flap, in combination with pedicle reliability, harvest convenience with no microsurgical requirements and minimal donor-site morbidity make them an ideal reconstruction choice for head and neck defects, especially during the COVID-19 pandemic.

The primary aim of our study is to present two cases of a total laryngectomy with revision of the pharyngoplasty and reconstruction of the posterior wall of the pharynx with the supraclavicular artery island flap during the COVID-19 period. The secondary aim is to review the literature on the use of SCAIF for pharynx reconstruction (surgical techniques, complications, contraindications).

## 2. Materials and Methods

From February 2020 to September 2021, two patients underwent reconstruction after resected tumors of the larynx and were included in the study. The supraclavicular artery island flap was used in both patients. Two surgeons and two assistants were involved in all cases. For the review, a manual search was conducted by one author (E.N.) in January 2022 in the PubMed, ScienceDirect, Wiley Online Library, Google Scholar, Scopus and Cochrane Library databases. Using MeSH terms Pharynx AND reconstruction AND flap, 412 articles were identified in the period 2000–2021 (Table 1). All types of articles in the English language, referring to adult patients with any type of laryngopharyngeal cancer requiring reconstruction with the SCAIF, were included. After the removal of duplicates, articles were evaluated. Pediatric and trauma patients, book chapters, anatomical studies and descriptions of other flap reconstruction techniques or in other anatomical regions were not included. Lastly, the abstracts of these articles were reviewed and evaluated according to inclusion and exclusion criteria. Titles and abstracts were assessed by two independent reviewers. Discussion solved any disagreements (E.N. and A.S.). PRISMA guidelines were applied in order to ensure the evidence-based process [8]. Critical Appraisal Skills Program (CASP) checklists evaluated studies for their validity and results [9]. Repeatability of the search results of Google Scholar is not possible [10].

## 3. Results

### 3.1. Case Reports

#### 3.1.1. Case Report 1

A 72-year-old male patient was admitted to the Ear–Nose–Throat (ENT) Department of Papanikolaou Hospital due to T3N0 squamous cell carcinoma of the larynx after adjuvant radiation therapy. He underwent a total laryngectomy and pharyngoplasty. A pharyngeal defect measuring 5 × 4 cm was reconstructed using a pedicled SCAIF. The donor site was closed primarily. The patient was discharged on postoperative Day 10. At the 1-year follow-up, the patient had the normal ability to swallow.

#### 3.1.2. Case Report 2

A 61-year-old male patient presented to our clinic with a squamous cell carcinoma (SCC) of the larynx. He had previous radiotherapy. A total laryngectomy and pharyngoplasty with one-stage reconstruction using the SCAIF was planned (Figure 1). The cutaneous part of the flap was used to create the neopharynx. At postoperative Day 15, a fasciocutaneous fistula was noticed and treated conservatively. No secondary surgery was required. At the one-year follow-up, functional and aesthetic results both of donor and recipient sites were assessed as satisfactory (Figure 2).

### 3.2. Review Results

The type of each study, date of publication, aim, conclusion endpoints and level of evidence were data extracted and presented. Ten full-text articles comprising 92 patients with 93 supraclavicular flaps were included according to the eligibility criteria [11,12,13,14,15,16,17,18,19,20]. The procedure is presented in Figure 3 (PRISMA 2020 flow diagram) [8].

#### 3.2.1. Study Characteristics

From the critical appraisal of the ten studies, there were no randomized controlled trials identified. Four studies were conducted in the USA [11,12,13,19], two in Europe [15,16], one in India [17], one in Iran [20] and one in China [18]. All studies were single-centered [9,10,11,12,13,14,15,16,21,22]. The eligibility criteria were clearly reported, increasing the applicability of the results (Table 2).

#### 3.2.2. Participant’s Characteristics

The review included 92 patients. A total of 93 supraclavicular flaps were used for reconstruction. Patients had undergone either partial or total laryngopharyngectomy. Different surgical techniques were used for the reconstruction, according to the type of defect: circumferential of the anterior wall of the pharynx. The mean flap size from the given data was 7 × 13.5 cm, and 71 of 92 patients had previous radio or chemoradiotherapy [11,12,13,14,15,16,17,21,22]. Regarding complications, 1 partial [15] and 1 near-total flap loss [13] were noticed, as well as 26 pharyngocutaneous fistulas [11,12,13,14,15,16,17,19,21,22], 4 postoperative stenosis [11,14], 1 seroma [16] and 1 incision dehiscence [13]. All donor sites were closed primarily (Table 2).

## 4. Discussion

### 4.1. Surgical Technique

The supraclavicular artery island flap is a versatile flap. Lee and Hwang assessed that the thickness of the supraclavicular fossa and adjacent skin over the upper half of the deltoid in the cervicohumeral area is approximately 1 mm thinner than neck skin, making it suitable for regional reconstruction [19]. Furthermore, its length, with a mean of 21.2 cm [18], and its width, with a mean of 9.7 cm [23], allow its proper configuration according to the type of laryngopharyngectomy. It can be used as a patch graft, which is the reconstruction of the anterior pharyngeal wall using the cutaneous portion of the flap, as a mucosal lining for total laryngectomy and partial pharyngectomy, or as a pharyngeal interposition graft, which is an onlay flap after primary closure of the pharynx [13]. For partial laryngopharyngeal defects, Liu and Chiu used the patch graft technique, with the proximal portion of the flap being de-epithelized and buried in the neck soft tissue [12]. Kucur et al. used the contralateral supraclavicular artery island flap as a patch graft for a left defect after partial laryngopharyngectomy due to previous ipsilateral neck dissection with intraoperative radiation therapy, with good results [14]. The reconstruction for circumferential pharyngeal defects is more demanding. Different techniques have been described. Tubularization of the supraclavicular artery pedicled flap is the most common technique, in which the neopharynx is totally created by the cutaneous part of the flap [16,21]. A modification of this technique is the “end-to-side” hypopharyngeal reconstruction for laryngeal-preserving partial pharyngectomy. In this technique, the distal part of the rolled flap, an 8 cm tube, is linked to the proximal part of the cervical esophagus, and the other end of the tubularized flap is attached to the lateral side of the oropharyngeal and postcricoid mucosa, applying the “end-to-side” technique [20]. On the contrary, the U-shaped supraclavicular pedicle flap for circumferential defects uses the lateral edges of the flap secured to the lateral prevertebral fascia and the distal end of the flap secured to the base of the tongue of the pharynx, giving a final “U” shape [16]. With this technique, the neopharynx is created with the anterolateral 270° rotation of the pedicle supraclavicular flap, and the posterior wall is made of the prevertebral fascia [16]. The partially tubularized flap has the advantage of a less wide flap, approximately 6–7 cm, given the opportunity of direct closure of the donor side, without the use of a skin graft. Finally, its color match to the neck area provides an excellent coverage option for anterior neck skin defects. Zhou et al. described the bilateral use of supraclavicular artery island flaps, one for the mucosal defect and the other to repair the anterior cervical skin defect [18].

### 4.2. Complications

#### 4.2.1. Pharyngocutaneous Fistulas

The most frequent immediate postoperative complicatio after pharynx reconstruction is the formation of pharyngocutaneous fistulas, reported in 5–65% of the literature [24]. This complication leads to prolonged hospital stay, delay of per os nutrition, gastric tube dependency and delayed voice restoration. In serious situations, additional surgical procedures are necessary. In our review, 26 pharyngocutaneous fistulas were reported in 92 patients. More specifically, all the included studies, except for two [18,20], had at least one postoperative pharyngocutaneous fistula. The majority of the patients with fistulas had a history of previous radiation or chemotherapy and radiotherapy combined. Reiter et al. compared the supraclavicular flap with the radial forearm flap and the anterolateral thigh flap for laryngeal reconstruction and concluded that this kind of flap used seems to be of minor importance since the pharyngocutaneous fistula formation rates are comparable [15]. Furthermore, prior radiotherapy and/or chemotherapy are independent risk factors for fistula formation, a finding that is consistent with those reported in previous studies [24]. The most important finding regarding fistulas after the SCAIF is that they tend to heal by secondary intention. Chiu et al. had 30% of early pharyngeal leaks, attributed to the flap’s input technique [11]. All six incidents were resolved spontaneously with a nothing-by-mouth diet [11]. On the other hand, in the study by Escalante et al, 38.9% developed fistulas, and 42.9% of them needed a secondary surgical intervention with another flap reconstructive option [19]. The main difference in this study is that the flap was used for total laryngectomy defects, and these patients with flap reconstruction had previous chemotherapy [19]. Overall, only five patients from the included studies needed revision surgery for pharyngocutaneous fistula formation.

#### 4.2.2. Flap Necrosis

One partial and one near-complete flap loss were noticed in 93 SCAIFs [13,15]. Partial necrosis was noticed at the distal part of the flap [15], whereas near-total flap loss was associated with pharyngocutaneous fistula [13]. Although the SAI flap is an axial pattern flap, it seems that there is a significant correlation between a flap longer than 22 cm and flap necrosis (*p* = 0.02) since the vascularity of the distal tip of the flap depends on the perfusion pressure of what is a relatively small-diameter artery [25]. The meticulous harvest of the flap and taking care of the vessels are essential during flap rising. However, when the SCAIF is used for laryngopharyngeal defects, the embedding of the flap in the anatomical position of the larynx makes visual monitoring of the flap impossible. Usually, postoperatively handled Doppler is used at the pivot point of the flap, accessing the blood flow of the main vessels. Other studies using the SCAIF for reconstruction of head and neck oncologic defects had partial flap necrosis rates of 4.2% to 14.9% and complete flap necrosis rates of 0% to 5.6% [25]. Furthermore, flap loss rates are comparable with those of the radial forearm flap (RFF) and the anterolateral thigh (ALT) flap. Reiter et al. compared the ALT flap, RFF and SCAIF as reconstructive options for laryngopharyngeal defects [15]. Regarding flap loss rates, they remained low in their cohort (3% complete flap loss for the RFF, 5% partial loss for the ALT flap and 6% partial loss for the SCAIF) [15]. Our results are comparable to other series published on pharyngeal reconstruction, making the supraclavicular flap a reliable option for laryngopharyngeal reconstruction [15,21,22].

#### 4.2.3. Stenosis

Unlike previous complications, stenosis of the neopharynx is a constant long-term complication of such surgeries, with a great impact on patient quality of life. The presence of stenosis leads to longer gastric tube dependency and may require dilatation therapies with a long recovery time. In our review, four stenoses were noticed [11,15]. This incidence is less than the average 14% of stenosis incidence mentioned in the available literature [26,27]. Ιn fact, stenosis is mostly seen in circumferential reconstruction techniques and in patients with prior radiotherapy/chemotherapy. Compared with other flaps, reconstruction with the supraclavicular flap is an equal alternative since no difference is noticed in stenosis rates. On the contrary, primary closure has higher pharyngeal strictures than flap reconstruction [28].

#### 4.2.4. Minor Complications

Minor complications, such as incision dehiscence [13], seroma [16] or topic cutaneous infections of the flap are treated conservatively. An interesting minor complication is a sensation in the shoulder when swallowing, mentioned by Chiu et al. [11] Obviously, this condition is related to the preservation of the nerves in the flap area. Τhis phenomenon is reduced as patients get used to their new condition [11].

#### 4.2.5. Donor Site

Undoubtedly, donor-site morbidity significantly influences patient satisfaction and quality of life following reconstructive surgery. When the supraclavicular flap is harvested up to 8 cm, primary closure of the donor site is performed, and no malformation is noticed. On the contrary, when the free RFF is used, a skin graft is needed for the donor site. Wide scars are a common donor-site complication after supraclavicular flap due to tension suturing with good follow-up results and no displeased patients [12]. Minor complications such as dehiscence and seroma of topic cutaneous infections may appear, but they can be treated topically by local wound care.

### 4.3. Contraindications

The SCAIF is based on the supraclavicular artery, which originates from the transverse cervical artery. Any suspicion of lack of pedicle integrity is the absolute contraindication for flap harvesting. This may occur in radical neck dissection due to oncological reasons, especially in level IV–V neck dissection, with the ligation of the transverse cervical artery. A duplex ultrasound examination reveals pedicle integrity. In special cases, a preoperative CT angiography can eliminate any doubts. However, even if the pedicle is preserved, previous elevation of the skin flap in the supraclavicular triangle may damage skin perforators. Therefore, during preoperative evaluation, Doppler should be applied at the course of the vessel until the acromioclavicular joint. Other types of surgery that may affect the pedicle are those for orthopedic reasons and should be investigated during patient medical history and clinical evaluation.

Regarding radiotherapy, it is considered a possible cause of pedicle compromise due to thrombosis. From our review, 71 out of 92 patients had undergone previous radiotherapy or chemotherapy before surgery. After pharynx reconstruction with the SCAIF, these patients confronted more complications. Two of them had partial and near-complete flap loss, whereas no such complication was noticed in non-radioexposed patients. Our results are in accordance with the results of Reiter and Baumeister, who found that preoperative radiotherapy led to higher rates of flap loss [15]. On the contrary, in a recent study evaluating the safety of supraclavicular flap in the setting of neck dissection and previous radiation therapy, the authors concluded that there was no association between prior neck dissection or radiation treatment and flap loss [29]. Furthermore, four patients with preoperative radiotherapy had to deal with neopharynx stenosis. Finally, patients both exposed and nonexposed to radiotherapy and/or chemotherapy appeared with pharyngocutaneous fistulas. It is obvious that even though patients with previous radiotherapy have a higher risk of complications, the supraclavicular flap is still a reliable flap option. If there is such an option, it is preferable to harvest the flap from a side without previous intervention [30].

### 4.4. Reconstruction with RRF and ALT Free Flaps

The SCAIF has similar characteristics to the free RFF: they are both thin, pliable and easy-to-harvest flaps. On the other hand, the ALT flap is thick, especially in obese patients, making difficult its incorporation to head and neck deficits.

Regarding this review, two of the included studies compared larynx reconstruction using both the local SCAIF, pectoralis flap [19] and free flaps, i.e., radial forearm flap and anterolateral thigh flap [15]. Escalante et al. concluded that the RFF is an excellent option for larynx reconstruction due to diminished length of hospital stay, significantly fewer fistula formation and lower gastric tube dependency. On the other hand, Escalante et al. support that the SCAIF is an equivalent alternative to larynx reconstruction compared to the RRF and the ALT flap since complications, flap loss rates and fistula formation rates, are similar for the three mentioned flaps [19].

As long as surgical technique is concerned, the SCAIF reduces operative time when compared with free-flap reconstruction, as it can be harvested in less time, and there is no need for microsurgical anastomosis [31]. When a two-team approach is used, one team for the flap harvesting and one team for the oncological resection of the tumor, there is no significant difference in operative time [15]. Usually, patients with free-flap reconstruction have a longer length of hospital stay compared with regional flap options due to postoperatively ICU stay, longer monitoring of the flap and recovery of the donor site [31]. One of the major findings of a recent systematic review, comparing free versus pedicled flaps for reconstruction of head and neck cancer defects, was that free flaps were associated with a lower hospitalization stay compared to pectoralis major pedicled flaps but a higher hospitalization stay when compared to the SCAIF, emphasizing the superiority of the SCAIF regarding hospitalization not only to free flaps, but also to some pedicle flaps [32]. Finally, the choice of the flap is always a personalized decision. A voluminous patient is not preferred for ALT flaps since there will be a bulky free flap. Therefore, a positive Allen test makes the RFF a contraindication for reconstruction. In addition, microsurgery requires advanced surgical skills and well-organized settings, which are not always available. When these conditions are not met, the supraclavicular flap is a reliable option.

### 4.5. COVID-19 Area and Use of SCAIF

During the ongoing COVID-19 pandemic, surgical practices have changed, and the use of local or regional flaps, instead of free flaps, has become a necessity. Before the COVID-19 pandemic, the SCAIF was an option for patients who were not elective for microsurgical reconstruction. The restrictions of the period we are going through have led to an increase in the use of the supraclavicular flap and in identifying its particular positive characteristics.

Our surgical team was confronted with two cases of laryngeal defects, which were successfully reconstructed using the supraclavicular flap. Due to preoperative radiotherapy, a fasciocutaneous fistula occurred in one patient, but it was managed conservatively. The initial surgical strategy was to use a free RFF for Case 1, but the lack of an ICU unit and reduced surgical time led to the modification of our initial plan. Finally, both cases had acceptable aesthetic and functional results. During this time period of the COVID-19 pandemic, 3 articles consisting of 37 cases of head and neck cancers were treated with pedicled flaps, including the supraclavicular flap (1 case report of oral SCC and synchronous esophageal cancer is reported [33], 31 cases of head and neck cancers treated with pedicle flaps [34] and 5 head and neck cases with supraclavicular flap) [35]. It is well mentioned that prior to the pandemic situation, the same patients would have potentially undergone primary reconstruction with a soft-tissue free flap [34], but satisfactory results with the supraclavicular flap change the workhorse in reconstructive surgery.

### 4.6. Limitations of the Study

Nonrandomized controlled studies on the SCAIF for pharynx reconstruction were found. Cohort studies, case series and case reports were used. Most importantly, case series and reports are uncontrolled, but this is not underpinning our research. As a result, they can suggest hypotheses but they cannot establish robust associations. Nevertheless, our findings are limited by the quality of the reported data. Since no randomization was used regarding the reconstructive flap options, patient selection bias is a fact. Furthermore, the variety of surgical techniques may affect the reported results. Prospective trials may demonstrate the real impact of the SCAIF on pharynx reconstruction.

## 5. Conclusions

The SCAIF is a reliable, easy-to-harvest and versatile fasciocutaneous flap that can be used for larynx reconstruction. Instead of free flaps, this flap requires no microsurgical technique, reduced operating time and reduced postoperative care. These advantages make the SCAIF an ideal option for reconstruction, especially for inexperienced surgeons in nonsupportive medical departments. Finally, this review underlined that the SCIAF has comparable results with other reconstructive options, consolidating this flap in the workhorse of laryngeal reconstructive surgery.

## Figures and Tables

**Figure 1 jcm-11-03126-f001:**
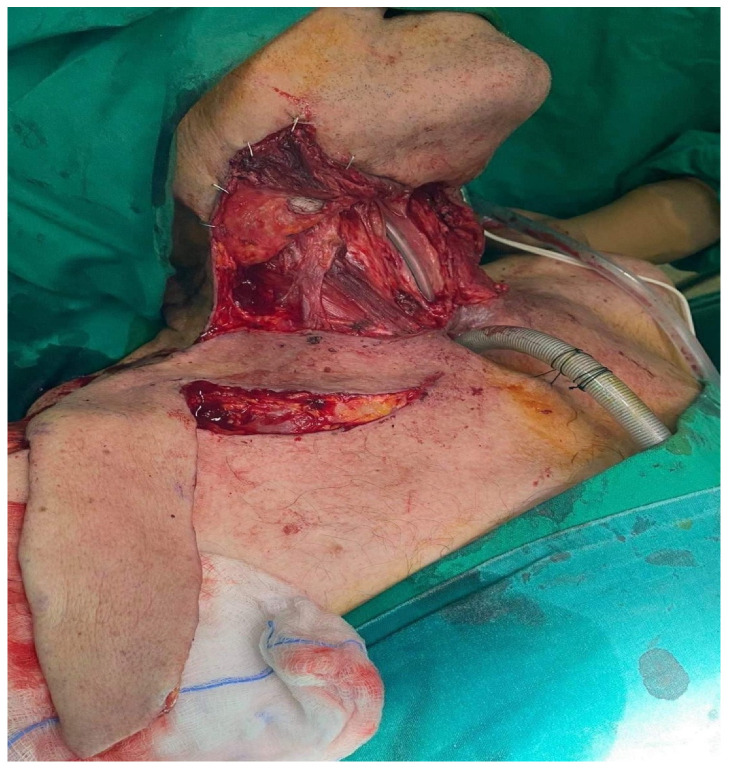
Intraoperative view of the defect and the supraclavicular flap raise.

**Figure 2 jcm-11-03126-f002:**
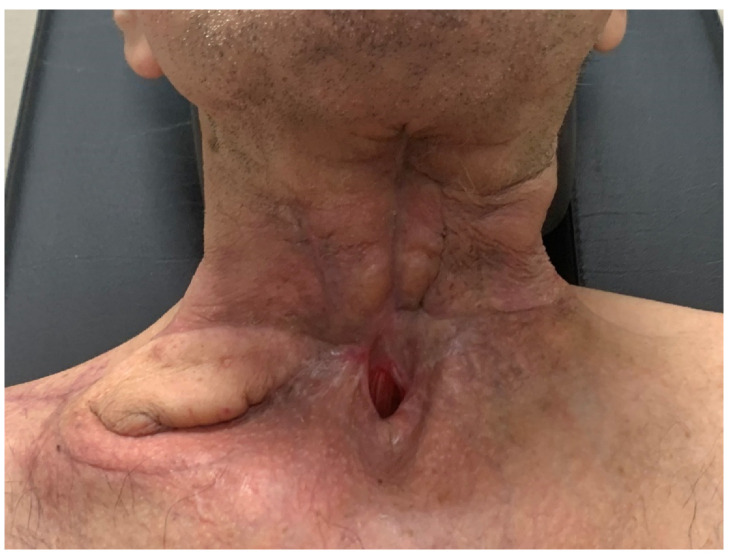
Aesthetic result at the 1-year follow-up.

**Figure 3 jcm-11-03126-f003:**
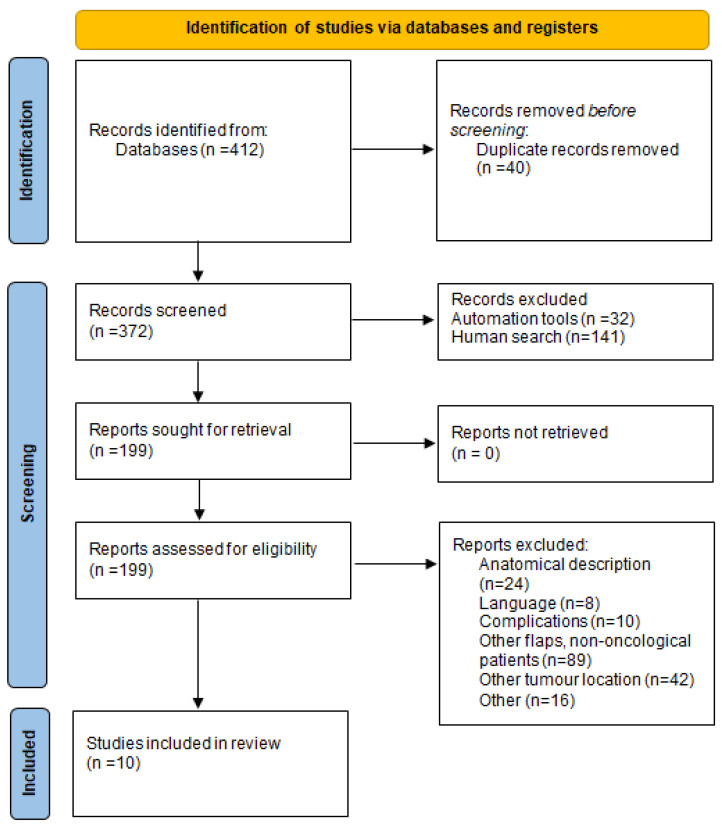
Prisma 2020 flow diagram [8].

**Table 1 jcm-11-03126-t001:** Search strategy used for each database with the corresponding results.

Electronic Databases	Search Strategy	Limits	Hits
Pubmed	Pharynx reconstruction AND flap (MeSH Terms)	Publication date: 1 January 2000–31 December 2021 English language	345
ScienceDirect	Pharynx reconstruction AND flap (MeSH Terms)	Year 2000–2021 Limited to title, abstract, or author-specified keywords	44
Wiley Online Library	Pharynx reconstruction AND flap (MeSH Terms)	Year: January 2000–December 2021 Publication type Journals Limited to title, abstract, or keywords	1
Google Scholar	Pharynx reconstruction AND flap (MeSH Terms)	Publication date 2000–2021 Limit to title	21
Scopus	Pharynx reconstruction AND flap (MeSH Terms)	Publication date: 2000–2021 English language Limited to title, abstract, or keywords	1
Cochrane Library	Pharynx reconstruction AND flap (MeSH Terms)	Publication date: 2000–2021	0
			412 Total hits

**Table 2 jcm-11-03126-t002:** Summary of included articles. (N = total number of patients, n = number of patients, SCAIF = supraclavicular artery island flap).

Article	Study Type, Setting	N	Defect	Type of SCAIF	Radio/Chemotherapy Prior Surgery (n/N)	Flap Size (cm)	Complications (n)	Revision Surgery
Chiu et al. (2008) [11]	Retrospective, New Orleans, LA	20	Partial or total laryngopharyngectomy	Circumferential pedicled flap	18/20	7 × 19.5	Leak—Pharyngocutaneous fistula (6) Sensation to the shoulder when swallowing (4) Strictures—stenosis (2)	-
Liu and Chiu (2009) [12]	Case series, New Orleans, LA	6	Partial pharyngectomy	Pedicled	6/6	6.5 × 20	Small leak—Pharyngocutaneous fistula (2)	-
Emerick et al. (2014) [13]	Retrospective, Boston, USA	15	Total laryngectomy	Pedicled	12/15	5.8 × 8.3	Pharyngocutaneous fistula (3, Radiation +) Near-complete flap loss (1, Radiation +) Incision dehiscence (1, Radiation −)	1
Kucur et al. (2015) [14]	Retrospective	1	Partial laryngopharyngectomy	Pedicled contralateral	1/1		Pharyngocutaneous fistula (1, Radiation +)	-
Reiter et al. (2018) [15]	Retrospective, Germany	18	Partial or total laryngopharyngectomy		7/18		Partial flap loss (1, Radiation +) Pharyngocutaneous fistula (2, 1 Radiation +) Stenosis (2, Radiation +) Gastric tube dependency (5, 2 Radiation +)	1
Carnevale et al. (2019) [16]	Retrospective, Spain	5	Total laryngectomy	U-shaped	3/5	7.5 × 20	Pharyngocutaneous fistula (1) Seroma (1)	-
Jonnalagadda (2019) [17]	Retrospective, India	7	Total laryngectomy	Circumferential	5/7	5.5 × 9	Pharyngocutaneous fistula (4)	1
Zhou et al. (2020) [18]	Retrospective, China	1	Partial laryngopharyngectomy	Bilateral	1/1	7 × 4 and 10 × 7	-	
Escalante et al. (2021) [19]	Retrospective, USA	18	Total laryngectomy		18/18		Pharyngocutaneous fistula (7)	3
Ahmadi et al. (2021) [20]	Case report, Iran	1	Partial laryngopharyngectomy	Tubed, end-to-side technique	0/1		-	
Total	10 articles	92			71/92		Pharyngocutaneous fistula (26) Flap loss (2) Stenosis (4)	6

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
