# Peer review of "The Supraclavicular Artery Island Flap for Pharynx Reconstruction"

_jcm, 2022, doi:10.3390/jcm11113126_

Round 1

Reviewer 1 Report

The paper is well written

However a bit of deficiency which can be rectified is as follows

1. Critical discussion of the reviewed cases

2. limitation of cases shown

3. A bit more on advantages and disadvantages of this island flap over the others.

Reviewer 2 Report

The manuscript entitled “The Supraclavicular Artery Island Flap for Pharynx Reconstruction” has useful information for readers who are interesting in this field. However, there are some problems to be solved.

1. Line 76,  “pharyncoplasty” should be changed to “pharyngoplasty”

2. Line 131-132,

The authors mentioned “At the one-year follow-up, functional and aesthetic results both of donor and recipient were assessed satisfactory.” I think they should present postoperative endoscopic photos or images of videofluoroscopy of neo-pharynx and postoperative photos of the wound.

3. Line 143-146,

These sentences were already described in the Material and Methods section. I think the authors should delete them in the Results section.

4. Line 163-164,

“8-16”, “12”, and “10” should be “[8-16]”, “[12], and “[10]”.

5. Line 198,

“[8, 14]” might be “[14, 18]”. Please confirm it.

6. Line 205,

“13” should be “[13]”.

7. Line 217,

The paper of reference No. 14 is not about this technique. Please make sure that the citation numbers of all reference papers are correct.

8. Line 206,

What is “207o”?

9. Line 223,

“wit” should be “with”.     

Author Response

Please see the attachment. We would like to thank the reviewer for his valuable comments.

Reviewer 3 Report

Dear Authors,

I perused your work. You raised a very useful topic especially in a period that required agility. However, I think you need to amend the structure of your draft. Please use the PRISMA 2020 guidelines. 

In terms of anatomical descriptions please use the paper: Kokot N, Mazhar K, Reder LS, Peng GL, Sinha UK. The Supraclavicular Artery Island Flap in Head and Neck Reconstruction: Applications and Limitations. JAMA Otolaryngol Head Neck Surg. 2013;139(11):1247–1255. doi:10.1001/jamaoto.2013.5057

Please avoid presenting the surgical anatomy in introduction and discussion. 

Your discussion covers interesting topics but without further analysing (ie percentages of fistula, flap necrosis among the quoted papers). 

You comments on comparison between free flaps and SCA flap, please provide evidence (papers, or reviews)

Please comment on the level of evidence based on the data you gleaned, strengths and drawbacks. You have included for example cases reports, this is not underpinning your research. Maybe data regarding QOL should be added if possible.

Reviewer 4 Report

The authors reported two cases of the SCAIF and conducted a systematic review to review the outcomes of this technique. Meta-analysis or comparison with other techniques was not carried out. There are several concerns which have be addressed.

The old PRISMA 2009 guideline was used. Please use the updated PRISMA (Page et al. 2021) instead. Figure 2 also has to be changed according to the new PRISMA. Cursor and underlines are still visible in Figure 2. Please export the figure as vector. 

Google scholar was used as one of the databases. But the search results yielded only 21 hits. Please make sure if this is correct. I tried to reproduce the results using the keywords “pharynx reconstruction” AND “flap” (2000-2021), and I found 111 entries. Please also acknowledge in the discussion that repeatability of the search results of Google Scholar is not possible (Bramer 2016 http://jmla.pitt.edu/ojs/jmla/article/view/61). 

The third column of Table 1 might be a mistake. It contains the word ‘AND’ in every single row. This column may be removed entirely. 

In the future, I recommend the authors to register their systematic review on the PROSPERO database before the review is conducted. This is to make sure that there is no similar ongoing review.

The whole manuscript should be checked for language errors. 

Minor remarks

- Line 119: The abbreviation ENT is used for the first time. Please spell it out.

- Line 163-164: 8-16, partial12, loss10. What are these? These mistakes can be seen throughout the manuscript. 

- Line 342: The abbreviation SCC has to be spelled out too.

- There are so many variants of the word COVID-19 such as “covid19”, “COVID19”, “COVID - 19”, “COVID- 19”, etc. Make them consistent!

Author Response

Please see the attachment. We would like to thank the reviewer for his valuable comment

Round 2

Reviewer 2 Report

Thank you for revision.

Author Response

Dear Reviewer 2,

I would like to thank you once more for your valuable comments.

Reviewer 3 Report

Thanking the authors for editing their manuscript.

Author Response

Dear Reviewer 3,

we would like to thank you for you suggestions and comments

Reviewer 4 Report

I thank the authors for their careful revision. The manuscript is much improved. However, there are still some issues with Figure 3. 

Image quality of the PRISMA chart is still limited. The underlines are still visible. The following texts in the upper right box have to be remove: "Records marked as ineligible..." and "Records removed for other reasons...". 

The authors also have to crop out the text at the top and the bottom.

To create underlines free text, the authors have to export the .docx containing the PRISMA flowchart into PDF and then open the PDF with software like Acrobat or Preview (if you use Mac) etc. Then a screenshot can be made.  If the authors are unable to do this, they can send me the .docx of the PRISMA flow chart. I am happy to help!

Author Response

Dear Reviwer 4,

We appreciate your willing to help. Your comments were more than valuable to us. Thank you for your suggestions. 
